# Metabolic Reprogramming toward Aerobic Glycolysis and the Gut Microbiota Involved in the Brain Amyloid Pathology

**DOI:** 10.3390/biology12081081

**Published:** 2023-08-03

**Authors:** Toshiyuki Murai, Satoru Matsuda

**Affiliations:** 1Graduate School of Medicine, Osaka University, 2-2 Yamada-oka, Suita 565-0871, Japan; pi3kp10@outlook.jp; 2Department of Food Science and Nutrition, Nara Women’s University, Kita-Uoya Nishimachi, Nara 630-8506, Japan

**Keywords:** glucose metabolism, aerobic glycolysis, Warburg effect, Alzheimer’s disease, amyloid-β, ketogenic diet

## Abstract

**Simple Summary:**

Amyloid-β toxicity is attributed to oligomer and fibrillar amyloid-β, which either damages the neurons or initiates an intracellular signaling cascade toward neuronal cell death. In addition to the amyloid-β aggregation, an emerging hallmark of Alzheimer’s disease is the metabolic reprogramming toward aerobic glycolysis. This article describes the emerging features of Alzheimer’s disease from the aspect of alternation in the central metabolic pathways in the human brain. The potential practical approaches for therapeutic intervention for the improvement of metabolic disorders are also described. The dietary intervention such as the use of ketone-mediated nutritional therapeutics is a promising strategy for complementing the brain energy levels in patients with Alzheimer’s disease as a high-ketogenic diet may restore the metabolic imbalance caused by aerobic glycolysis in the neurons and microglia.

**Abstract:**

Alzheimer’s disease (AD) is characterized by the formation of senile plaques consisting of fibrillated amyloid-β (Aβ), dystrophic neurites, and the neurofibrillary tangles of tau. The oligomers/fibrillar Aβ damages the neurons or initiates an intracellular signaling cascade for neuronal cell death leading to Aβ toxicity. The Aβ is a 4 kDa molecular weight peptide originating from the C-terminal region of the amyloid precursor protein via proteolytic cleavage. Apart from the typical AD hallmarks, certain deficits in metabolic alterations have been identified. This study describes the emerging features of AD from the aspect of metabolic reprogramming in the main pathway of carbohydrate metabolism in the human brain. Particularly, the neurons in patients with AD favor glycolysis despite a normal mitochondrial function indicating a Warburg-like effect. In addition, certain dietary patterns are well known for their properties in preventing AD. Among those, a ketogenic diet may substantially improve the symptoms of AD. An effective therapeutic method for the treatment, mitigation, and prevention of AD has not yet been established. Therefore, the researchers pursue the development and establishment of novel therapies effective in suppressing AD symptoms and the elucidation of their underlying protective mechanisms against neurodegeneration aiming for AD therapy in the near future.

## 1. Introduction

Neurodegenerative diseases including Alzheimer’s disease (AD) have been affecting public health worldwide, and are estimated to increase three-fold by 2050 [1,2]. The well-characterized biomarker of neurodegeneration is the accumulation of misfolded and aggregated proteins in the brain. However, the definition of neurodegenerative diseases is descriptive, i.e., the formation of specific proteinaceous aggregates which is a key hallmark of a variety of neurodegenerative diseases and often serves for their diagnosis and pathological classification. AD is characterized by the senile plaques (SPs) consisting of fibrillated amyloid-β (Aβ) and dystrophic neurites, and the formation of neurofibrillary tangles (NFT) consisting of the hyperphosphorylated tau protein. It is hypothesized that the cellular phase of AD involves the aggregation of Aβ that acts as a trigger causing homeostatic imbalance and neuroinflammation [2]. Apart from the typical hallmarks of AD, i.e., aberrant synaptic integrities and progressive neuronal cell death [3], certain deficits in alternative splicing and metabolic alterations have also been identified [4]. Herein, the emerging features of AD from the aspect of alternation of the pivotal pathway of carbohydrate metabolism in the human brain are described. Possible dietary interventions for AD including ketone-mediated nutritional therapeutics are also discussed.

## 2. Amyloid Aggregation in Alzheimer’s Disease

AD is characterized by the deposition of specific peptides such as Aβ extracellularly and tau intracellularly in the brain [3]. Currently, the established biomarkers are Aβ_42_, total-tau, and phospho-tau in the cerebrospinal fluid (CSF). The amyloid hypothesis proposes Aβ accumulation to be the major cause of AD. Aβ is the main component of plaques and is derived from a cell surface transmembrane protein, the amyloid precursor protein (APP). It has an approximately 4000 molecular weight and originates from the C-terminal region of the APP via proteolytic cleavage. The proteolysis of the APP by an α-secretase liberates a soluble APP-α from the surface and leaves a C-terminal APP fragment at the cell surface. The amyloidogenic cleavage of APP is executed through sequential proteolysis by β-secretase and γ-secretase at the N-terminus and C-terminus of the APP, respectively. APP is produced in most human peripheral cells, and amyloid-β is detected in the cerebrospinal fluid and the blood plasma. There are two main isoforms of Aβ peptide in humans: Aβ_40_ and Aβ_42._ The Aβ_40_ is predominantly abundant, while Aβ_42_ can form fibrils more intensively and is considered comparatively more neurotoxic. Aβ is predominantly expressed at the cell membrane and translocated to the extracellular space, where it is deposited as SPs, which are characteristic features of AD. The toxicity of Aβ is attributed to oligomers/fibrillar Aβ, which either damage the neurons or initiate an intracellular signaling cascade causing neuronal cell death. The widely recognized model for AD progression suggests that Aβ pathogenesis may be an upstream event in AD and functions as a trigger of downstream pathways, including the tau-mediated toxicity, misfolding of hyperphosphorylated tau isoforms, tau accumulation in tangles, and proliferation of tau proteins leading to cortical neurodegeneration [5].

## 3. Metabolic Reprogramming in the Brain

About a hundred years ago, Otto H. Warburg found that tumor cells tend to utilize aerobic glycolysis to obtain energy and concomitantly produce a large amount of L-lactate [6]. This discovery termed the Warburg effect has been observed in various tumor types including glioblastoma [7]. The Embden–Meyerhof–Parnas pathway (EMP pathway), i.e., glycolysis, and Krebs cycle, also known as the tricarboxylic acid cycle (TCA cycle), are the central metabolic pathways providing biochemical precursors for biosynthesis and energy production (Figure 1). Most tumor cells prefer aerobic glycolysis to obtain massive energy, where a large fraction of pyruvate produced by glycolysis is converted into lactate [8]. Although certain genetic impairments in the functioning of the oxidative metabolism enhance aerobic glycolysis in tumors, aerobic glycolysis does not generally predict the loss of oxidative metabolism [9].

Recently, an accumulating body of evidence suggests that the Warburg effect not only occurs in the tumor cells but also in non-cancerous cells. This indicates that the role of the Warburg effect may go far beyond cancer progression, implying a critical role in non-cancer diseases and physiological activities. It has been reported that aerobic glycolysis and lactic acid production are principal features of astrocytes [10]. The astrocyte-neuron-lactate shuttle (ANLS) hypothesis suggests that the astrocyte-derived lactic acid is assimilated by the neurons and re-converted to pyruvic acid to drive the mitochondrial respiratory chain. Though the ANLS hypothesis is causing controversy, an apparent contradiction exists between the ANLS hypothesis and the Warburg effect involving enhanced glucose uptake by the neurons and release of L-lactate.

The human brain, albeit being only a fraction of the total body weight, accounts for one-fifth of the total energy utilization of the entire body at its resting state [10]. Two main types of cells exist in the brain—the neurons utilizing ~70% of the total brain energy, and glial cells consuming the remaining energy. The glucose consumption rate in the brain depends upon the type of the cell and the expression level of each enzyme. It has been thought that the neurons are metabolically oxidative, and astrocytes are relatively glycolytic [11]. A recent study indicated that the neurons utilize glucose predominantly through glycolysis in vivo and require it for normal functioning [12].

In neurons, glucose is metabolized through either the glycolysis or pentose phosphate pathway (PPP), followed by the Krebs cycle, and oxidative phosphorylation (OXPHOS) system, resulting in the production of water, carbon dioxide, and 30–36 adenosine triphosphate (ATP) molecules (Figure 1). The glycolysis process converts glucose to pyruvate, and then pyruvate is transported to the mitochondria in which pyruvate is used to synthesize acetyl coenzyme A (acetyl-CoA). Further, acetyl-CoA is combined with citrate which then enters into the TCA cycle. These pathways produce nicotinamide adenine dinucleotide (NADH) and flavin adenine dinucleotide (FADH_2_), which are subsequently re-oxidized in the electron transport chain (ETC) in the mitochondrial respiratory chain to generate ATP.

## 4. Metabolic Reprogramming and Alzheimer’s Disease

Metabolic programming is defined as the effects of an environmental event or condition (for example, energy restriction or exposure to cell stressors) that induce long-lasting effects on metabolism in progeny (organismal or cellular) and may even have sustained effects over multiple generations [13]. Metabolic rewiring is often used interchangeably with metabolic programming [13]. Currently, the deregulation of cellular energetics has been commonly termed “metabolic reprogramming” [14], which is frequently used to describe a common feature of cancers [15].

The human brain largely depends on glucose as its primary source of energy to fuel the respiratory chain in the mitochondria. Glucose metabolism is enhanced and partially utilized to provide the lipid molecules necessary for neurite growth [16]. The aerobic glycolysis in the brain might support neurite outgrowth and decreases with age, whose functional significance in the context to the development of AD symptoms remains to be elucidated. According to a recent study, the cognitively impaired status was related to a decrease of the typical pattern of aerobic glycolysis in young adults [17]. This indicates that aerobic glycolysis might be involved in the responses to the early phase of AD pathogenesis, and the age-associated white matter lesions might impair the processes.

With regard to AD, Traxler and colleagues identified the metabolic signature induced by pyruvate kinase M2 (PKM2), an isoform of pyruvate kinases, in the neurons (Figure 1). These neurons prefer glycolysis in spite of a normal mitochondrial function in a manner resembling the Warburg effect using a series of experimental techniques including multimodal-omics on AD patient-derived induced neurons [18]. Four pyruvate kinase isoforms are present in humans: the L and R isoforms are encoded by the PKLR gene, and expressed mainly in liver and red blood cells; PKM1 and PKM2 are encoded by the PKM gene, and expressed by alternative splicing [19]. PKM1 is ubiquitously expressed in adult differentiated tissues including the brain, while PKM2 is usually expressed predominantly in the fetus, adult stem cells, and neoplastic tissues [20]. Unlike pyruvate kinase M1 (PKM1), which forms a tetramer, PKM2 can be translocated into the cell nuclei and trigger the activation of transcription factors including the signal transducer and activator of transcription 3 (STAT3) and hypoxia-inducible factor 1α (HIF1α) to modulate the expression of effector molecules involved in apoptosis [21]. Thus, it is likely that transport of PKM2 to the cell nucleus does not lead to apoptosis via depletion of PKM2 tetramers from the cytosol, but rather functions in the formation of a nuclear apoptosis-inducing signaling complex [22]. The Warburg effect is associated with escaping apoptosis in many cancer types, while neurons enhance their ability to receive proapoptotic stimuli. The neuronal PKM2 isozyme induced a metabolic shift, which was partially inhibited by a naphthoquinone derivative “shikonin” isolated from the herb *Lithospermum erythrorhizon*, acting as a chemical inhibitor of PKM2 [17]. These results suggest the potential application of the drug-based targeting of PKM2 for the treatment and/or prevention of age-related neurodegeneration like in AD. Further, accumulating evidence suggests that activities of microglial cells relying on enhanced aerobic glycolysis markedly influence brain amyloid pathology [17]. A future study concerning the microglial–neuronal interactions might elucidate the molecular mechanisms underlying microglia-regulated neuron dysfunction in AD [23].

## 5. Oxidative Stress and Its Connection to Alzheimer’s Disease

Oxidative stress is an imbalance between the generation of reactive oxygen species (ROS) and the antioxidants and has been shown to contribute markedly to the pathogenesis and progression of AD [24]. The ROS represents reactive chemicals derived from molecule O_2_, which are continuously produced and catabolized in the cells of aerobic organisms. The free radical types of ROS include superoxide anion radicals and hydroxyl radicals; the non-free radical ROS forms include H_2_O_2_ and O_3_ [25]. Excessive ROS production participates in the pathogenesis of a variety of disorders; the dysregulation of ROS metabolism enhances the neurodegenerative diseases including AD.

ROS also play a role as secondary messengers in the redox-mediated signaling transduction pathways including the cascade of mitogen-activated protein kinases (MAPKs), p38 MAPKs, and c-*Jun* N-terminal kinases (JNKs). Therefore, ROS exhibits bivalent functions, with the other being the induction of the Akt pathway [26] through the inhibition of the counteracting phosphatase and tensin homolog (PTEN). The events triggered via the phosphatidylinositol-3 kinase (PI3K) pathway involve the activation of nicotinamide adenine dinucleotide phosphate (NADPH) oxidase and the production of ROS. Thus, ROS are crucial to intracellular redox regulation and positively regulate the PI3K signal pathway via mechanisms regulating the reversible oxidation and inactivation of PTEN and other enzymes that negatively regulate this pathway.

## 6. Microbiome-Microglia Interactions in Alzheimer’s Disease

The study of the gut microbiome is gaining prominence in the fields of biology and medicine and has particularly emerged as a key regulator of brain pathophysiology [27]. The maintenance of healthy gut microbiota is one of the critical factors for supporting the homeostasis of the immune system and cognitive–emotional balance via the production of several bioactive metabolites, thus giving rise to the gut microbiota–brain axis [10]. The composition of the gut microbiome is generally variable among individuals, and once the profile of the commensal microbiota is established to a certain extent during childhood, it exhibits strong resilience, i.e., its composition and activity subsequently remain substantially stable. The resilience of the health-promoting microbiota protects the host from a variety of dysbiosis-related pathologies. Thus, the interventions targeting them might be promising strategies for treating diseases and promoting health.

The mechanisms underlying the effects of the intestinal microbiome on the brain partially involve neuronal-immune associations [28]. The intestinal microbiome-derived metabolites are mainly generated in one of three ways: directly from ingested materials, from host-derived compounds, or de novo from primary metabolites. There is evidence for a strong causal link between small compounds produced by each of these scenarios and host health [28]. Although it is still not conclusive whether changes in the microbiome are consequences of or contributors to disease, evidence for the transmission of information between the gut and the brain, the gut−neuroimmune axis, is well established [28].

The microbiome-derived metabolites such as short-chain fatty acids (SCFAs), e.g., acetate, propionate, and butyrate are the commonly identified signaling metabolites that affect microglia. In addition to serving as energy sources for neurons and affecting the maturation of microglia, these SCFAs may influence the physiological function of the neurons. SCFAs are also able to regulate the levels of secretory neurotransmitters and neurotrophic factors. For example, acetate has previously been demonstrated to regulate the release of neurotransmitters including glutamate, glutamine, and γ-amino butyric acid in the hypothalamus and increase the neuropeptide expression. Propionate and butyrate exert a distinct influence on the intracellular K^+^ levels, implying the involvement of SCFAs in intracellular signaling regulation. Microglia receive the local environmental cues in the brain and respond to signals from the distal regions of the body, e.g., from the gastrointestinal (GI) tract. Therefore, the analysis of the physiological role of the GI microbiome in regulating the maturation of microglia and their proper function in the central nervous system is gaining momentum, because the gut microbe-derived SCFAs function as mediators of the gut–brain axis [29]. The brain has long been considered one of the immune privileged sites, but the concept has been challenged and revisited recently. Indeed, peripheral immune cells access the central nervous system. It is likely that SCFAs can also influence brain function through their interactions with the innate and adaptive immune system helping maintain homeostasis in the central nervous system, with dysfunctional regulation occurring in diseases including AD, which have been extensively reviewed elsewhere [30].

However, the molecular mechanism underlying this crosstalk remains largely uncovered. A recent study identified one of the microbiota-derived SCFAs, acetate, as a signaling metabolite that can enhance the maturation of microglia, maintain the homeostatic metabolic state, and regulate microglial phagocytosis and AD pathological progression during neurodegeneration [31] (Figure 2). This study further demonstrated that the supplementation of acetate to the diet of a mouse model for AD induced the pro-inflammatory phenotype of microglia with elevated cytokine expression, which was previously shown to suppress the phagocytosis by microglia in the presence of Aβ. These observations suggest that acetate is a crucial, GI microbe-derived signaling molecule that drives the metabolic pathways of microglia in AD by epigenetic modification. This study may pave the way for the development of microbiota-targeted therapeutics to modulate brain microglia in patients with AD [32].

## 7. Amyloid-β and Aerobic Glycolysis in Alzheimer’s Disease

It is important to note that the dysregulation of glucose assimilation in neurodegenerative diseases is not limited to the neurons but to the glial cells, particularly the astrocytes which can survive on aerobic glycolysis without mitochondrial respiration, and thereby might support neuronal physiological activities [33]. However, recently Aβ has shown induction of metabolism shift from OXPHOS to aerobic glycolysis in microglia [34] (Figure 2). Under such a situation, the two metabolic systems might be dysregulated, which results in microglial dysfunction. The mechanism behind this involved phosphorylation of the mammalian target of rapamycin (mTOR) protein and enhanced expression of HIF-1α in the microglia by Aβ, thereby activating the pro-inflammatory signals. The higher rate of mTOR and HIF-1α signals occurred with a reduction in O_2_ consumption and an enhancement in extracellular acidification, thus shifting the glycolytic balance in microglia [35]. This metabolic reprogramming of the microglia may drive Alzheimer’s pathogenesis, since enhanced generation of lactic acid results in regulations that epigenetically trigger glycolysis-associated gene expression via a positive loop. Further, amyloid triggers immunological tolerance and causes dysregulation in the glycolysis in the microglia. However, as stated, there are still several limitations [34]: The precise mechanism underlying the dysregulation of mTOR signaling and glycolysis in microglia remains to be fully elucidated. Future experiments will elucidate the mechanism underlying microglia’s defect, and further study on microglial metabolism may open the door to a better understanding of AD pathology and its therapy [34]. Overall, these findings suggest a significant neuro-immune dysregulation which can mediate the AD-associated pathologic conditions.

## 8. Nutrition-Based Interventions for the Control of Alzheimer’s Disease and Amyloid Pathology

An accumulating body of evidence indicates that dietary choices have a certain role in protecting against the neurodegeneration associated with AD, but the exact association between the nutritional profile of the diet and its neuroprotective effects remains largely unknown [36]. In addition, it is quite difficult to examine the distinct effects of each diet plan involved. Although a variety of lifestyle factors may influence the central nervous system (CNS), regulation of diet-related facets may be a possible method for the prevention of CNS dysfunction. The current understanding of the mechanisms involved in AD suggests the function of oxidative stress, which occurs when the generation of ROS exceeds the capability of the antioxidant-mediated sequestering system especially with aging, which may contribute to the progression of age-related neurodegeneration including AD. Thus, nutraceuticals with antioxidant activity, such as dietary plant polyphenols may benefit the prevention of AD [37,38] (Figure 2).

Certain dietary patterns in Japan are well known for their ability to prevent AD. The Ogimi diet rich in L-serine which is similar to the Okinawa diet, has been approved for Phase II clinical trials in patients of AD [39]. The ketogenic diet, assuming a very high-fat/low-carbohydrate, is aimed to trigger a metabolic reprogramming from the glucose metabolism towards the fatty acids metabolism yielding ketone bodies including β-hydroxybutyrate as substrates for energy. A high-ketogenic diet exhibited the potential for alleviating the symptoms of AD, suggesting the applicability of ketone-based nutritional therapeutics as a promising strategy for the complementation of the energy levels in the brain in patients with AD [40] (Figure 2). The Mediterranean diet has been associated with an improvement in memory, an increase in Aβ_42_:Aβ_40_, and pTau_181_ [41]. This association between the Mediterranean diet and a memory score is consistent with a previous study performed on a German DELCODE (DZNE-Longitudinal Cognitive Impairment and Dementia) project participants [42], and also with the view of the Mediterranean diet as a protective lifestyle factor against dementia [43]. Moreover, the association between the Mediterranean diet and Aβ_42_:Aβ_40_ but not with Aβ42 level is consistent with a previous observation that the Aβ42/40 ratio is a more sensitive biomarker for AD compared to Aβ42 [44].

In addition to these diets, various dietary patterns including Mediterranean-DASH Diet Intervention for Neurological Delay (MIND) are suggested to reduce the hallmarks of AD [45]. The relationship between various dietary patterns and AD development has been summarized in a systematic review of the literature [46]. Among these diets, the high-ketogenic diet is the most promising therapeutic method for mitigating AD because it may restore the metabolic imbalance caused due to the occurrence of aerobic glycolysis in the neurons and the microglia. The proposed high-ketogenic diet with low carbohydrates enhanced the supply of ketone bodies biosynthesized in the liver using free fatty acids as a substitute for those cells with a high demand for glucose (Figure 1). A well-formulated diet that is adequately ketogenic was regarded to be nutritionally safe, although prone to being deficient in micronutrients [47]. Additionally, in a recent meta-analysis, a healthy diet comprised of higher consumption of fruits, vegetables, legumes, and fish but a minimal intake of saturated fat and sodium, has been identified to demonstrate potential protective effects against AD. The plausible mechanisms through which a high-ketogenic diet could improve Aβ levels in AD are an alteration in the glycolytic metabolism, a decrease in the generation of ROS, assimilation of ketones as an alternative source of energy in the neurons, and amelioration of neuronal inflammation [48] (Figure 3).

## 9. Conclusions

The brain amyloid pathology is the major hallmark of AD, and the widely recognized model for AD progression suggests that Aβ pathogenesis may be an upstream event in AD and functions as a trigger of downstream pathways, including the tau-mediated toxicity, misfolding of hyperphosphorylated tau isoforms, tau accumulation in tangles, and proliferation of tau proteins leading to cortical neurodegeneration. In addition to these pathological features, an emerging hallmark of AD is the metabolic reprogramming toward aerobic glycolysis. Specifically, the neurons in patients with AD favor glycolysis despite a normal mitochondrial function indicating a Warburg-like effect. Given that oxidative stress and the gut microbiome are also implicated in the progression of AD in a complicated manner, the development and establishment of novel therapies effective in suppressing AD symptoms and the elucidation of their underlying protective mechanisms against neurodegeneration aiming for AD therapy are desired.

## 10. Future Directions

A therapeutic method that is significantly effective in the treatment, mitigation, and prevention of AD has not yet been established, while certain dietary patterns are well known for their properties in preventing AD. Among those, a ketogenic diet may substantially improve the symptoms of AD. A high-ketogenic diet may restore the metabolic imbalance caused by aerobic glycolysis in the neurons and microglia. Recent studies suggest that nutrition-based intervention targeting glucose metabolism may result in significant mitigation of Aβ-related AD pathology and neurodegeneration [40,46,47]. They also imply that the modulation of the intestinal microbiome may be a promising therapeutic option to prevent AD. The potentially practical approaches for the therapeutic intervention of metabolic disorders might be diet-based interventions. Such studies might lead to the establishment of novel therapeutic interventions for AD in the near future.

## Figures and Tables

**Figure 1 biology-12-01081-f001:**
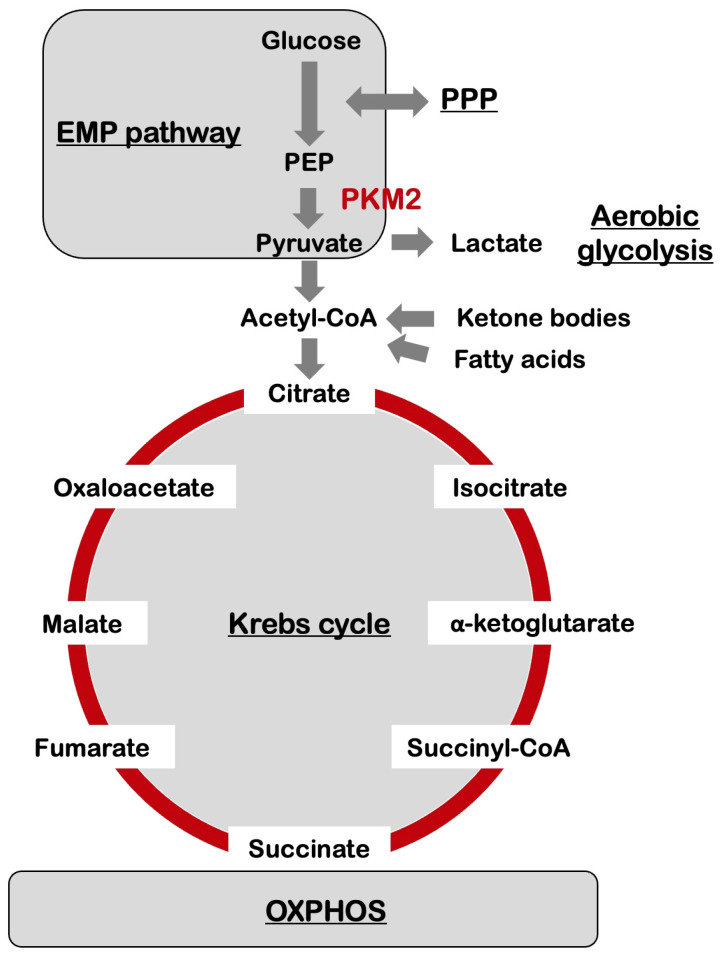
The central glucose metabolic pathways in cells of the nervous system. The Embden-Meyerhof-Parnas pathway (EMP pathway), i.e., glycolysis, and Krebs cycle, also known as the tricarboxylic acid cycle (TCA cycle), are the central metabolic pathways providing biochemical precursors for biosynthesis and energy production. In neurons, glucose is metabolized through either the glycolysis or pentose phosphate pathway (PPP), followed by the Krebs cycle, and oxidative phosphorylation (OXPHOS) system, resulting in the production of water, CO_2_, and 30–36 ATP molecules. The glycolysis process converts glucose to pyruvate, and then pyruvate is transported to the mitochondria in which pyruvate is used to synthesize acetyl-CoA. Further, acetyl-CoA is combined with citrate which then enters into the Krebs cycle. These pathways produce nicotinamide adenine dinucleotide (NADH) and flavin adenine dinucleotide (FADH_2_), which are subsequently re-oxidized in the electron transport chain in the mitochondrial respiratory chain to generate ATP. With regard to Alzheimer’s disease (AD), the metabolic signature induced by pyruvate kinase M2 (PKM2), an enzyme that converts phosphoenolpyruvate (PEP) to pyruvate, causes a metabolic rewiring towards glycolysis in a manner resembling the Warburg effect in which pyruvate produced by glycolysis is intensively converted into lactate. A high-ketogenic diet exhibited the potential for alleviating the symptoms of AD, suggesting the applicability of ketone-based nutritional therapeutics as a promising strategy for the complementation of the energy levels in the brain in patients with AD. The high-ketogenic diet with low carbohydrates enhanced the supply of ketone bodies biosynthesized in the liver using free fatty acids as a substitute for those cells with a high demand for glucose.

**Figure 2 biology-12-01081-f002:**
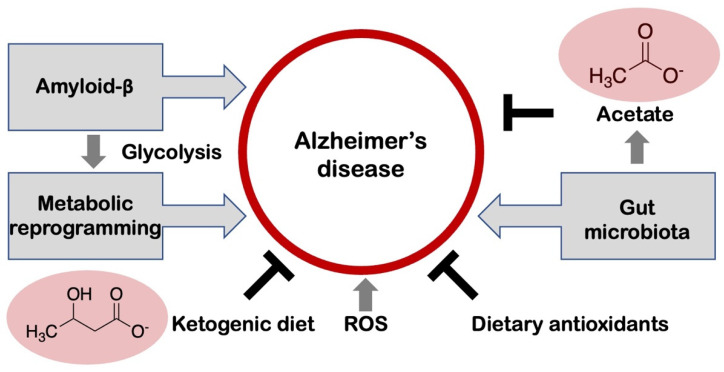
The multiple factors that may contribute to the pathogenesis and progression of Alzheimer’s disease. Alzheimer’s disease (AD) is characterized by the deposition of amyloid-β (Aβ) extracellularly in the brain. Aβ has shown induction of metabolic reprogramming from OXPHOS to aerobic glycolysis in microglia, which may drive AD pathology. Oxidative stress is an imbalance between the generation of reactive oxygen species (ROS) and the antioxidants and has been shown to contribute markedly to the pathogenesis and progression of AD. Gut microbiota-derived short-chain fatty acid, acetate, acts as a signaling metabolite that can enhance the maturation of microglia, maintain the homeostatic metabolic state, and regulate microglial phagocytosis and AD pathological progression during neurodegeneration. Nutraceuticals with antioxidant activity, such as dietary plant polyphenols may benefit the prevention of AD. The ketogenic diet yielding ketone bodies including β-hydroxybutyrate (the chemical structure is shown on the left of the figure) exhibited the potential for alleviating the symptoms of AD, suggesting the applicability of ketone-based nutritional therapeutics as a promising strategy for the complementation of the energy levels in the brain in patients with AD.

**Figure 3 biology-12-01081-f003:**
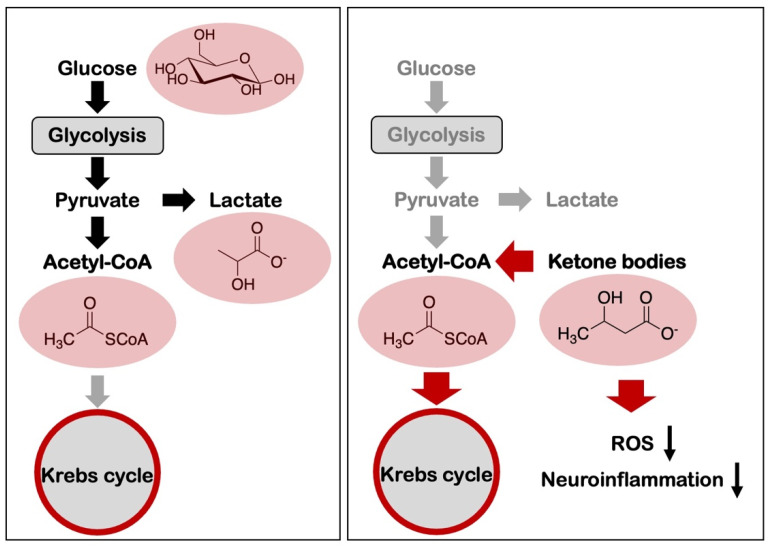
Potential mechanisms of high-ketogenic diet in brain amyloid pathology. The high-ketogenic diet generates ketone bodies that are metabolized to form acetyl-CoA, which could shift from the glycolytic metabolism requiring the contribution from glycolysis and lactate metabolism. The high-ketogenic diet could decrease the generation of ROS and ameliorate neuroinflammation.

## Data Availability

No new data were created or analyzed in this study. Data sharing is not applicable to this article.

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
