# Peer review of "Metabolic Reprogramming toward Aerobic Glycolysis and the Gut Microbiota Involved in the Brain Amyloid Pathology"

_biology, 2023, doi:10.3390/biology12081081_

Round 1

Reviewer 1 Report

Reviewer comments and suggestions

This study describes the emerging features of Alzheimer’s disease (AD) from the aspect of alternation in the pathway of carbohydrate metabolism in the human brain. The study suggested that the neurons in patients with AD favor glycolysis despite a normal mitochondrial function indicating a Warburg-like effect. Additionally, the authors discussed a ketogenic diet may substantially improve the symptoms of AD. The researchers in this study pursue the development and establishment of novel therapies effective in suppressing AD symptoms and the elucidation of their underlying protective mechanisms against neurodegeneration 

Overall, the manuscript was well written. However, a few concerns/comments needed to be explained/modified. 

  1. Comments for abstract: The authors at least describe the word metabolic reprogramming in the abstract part (in a single line)
  2. Line 75-78 The lines need references
  3. Section 4. It needs to define metabolic reprogramming in one paragraph
  4. Line 147-148 what does the line indicate?
  5. Line 156-159 The lines should be well explained
  6. Line 167 The authors did not discuss microglia in the previous section
  7. Line 216, reference 20 please discuss thoroughly.
  8. Line 229-231 The manuscript could be discussed more on this topic
  9. Line 236 comments for figure 2 It would be nice to draw some molecular figures that suit the standard of the journal
  10. Sections 7 and 8 both, The authors could enhance information related to this section
  11. Line 283 reference 32 Discuss the study thoroughly
  12. Line 311 First time used in the conclusion part (stem cell therapy) if the authors want to highlight the same, it should be well explained in the text
  13. Line 313 where was the recent studies, please read your manuscript carefully and cite relevant references 

Author Response

> Reviewer1 

> 1. Comments for abstract: The authors at least describe the word metabolic reprogramming in the abstract part (in a single line)

Following the advice, the word metabolic reprogramming has been added to the abstract part (page 1, line 25).

> 2. Line 75-78 The lines need references

Following the advice, a reference has been cited (page 2, line 76; reference no.5).

> 3. Section 4. It needs to define metabolic reprogramming in one paragraph

A paragraph has been added that describes the terminology of metabolic programming, metabolic rewiring, and metabolic reprogramming (page 4, lines 138-144).

> 4. Line 147-148 what does the line indicate?s

The sentence has been deleted to avoid confusion.

> 5. Line 156-159 The lines should be well explained

In accordance with the comment, the line has been further explained (page 4, lines 158-163).

> 6. Line 167 The authors did not discuss microglia in the previous section

A sentence describing the role of microglia has been added (page 5, lines 176-178).

> 7. Line 216, reference 20 please discuss thoroughly.

Sentences with further discussion have been added for the reference no.28 of the revised manuscript (page 6, lines 213-219).

> 8. Line 229-231 The manuscript could be discussed more on this topic

Following the advice, this topic has been discussed more deeply by adding a few sentences and citing a comprehensive review on SCFA’s effects on CNS (page 6, lines 234-240).

> 9. Line 236 comments for figure 2 It would be nice to draw some molecular figures that suit the standard of the journal

Accordingly, Figure 2 has been modified by adding the molecular figures.

> 10. Sections 7 and 8 both, The authors could enhance information related to this section

In accordance with the advice, enhanced information related to these sections has been added.

> 11. Line 283 reference 32 Discuss the study thoroughly

Following the advice, we deeply addressed the topic of this reference (the reference no.44 of the revised manuscript), Mediterranean diet, by adding sentences and new references (page 8, lines 316-322).

> 12. Line 311 First time used in the conclusion part (stem cell therapy) if the authors want to highlight the same, it should be well explained in the text

The sentence has been deleted to avoid confusion.

> 13. Line 313 where was the recent studies, please read your manuscript carefully and cite relevant references 

In response to the comment, we slightly modified the sentence and cited the relevant references (the references no.40, 46, and 47 of the revised manuscript).

Reviewer 2 Report

1. Which is the difference between simple summary and abstract?

2. Review does explain many topics but nothing deeply, this must be changed.

3. I think that Diet and AD prevention is a very hot topic, however is explained only shortly in this work.

4. It will be very interesting to inquiere between other countrys and AD.

5. A figure is missing concerning this paragraph: The plausible 293 mechanisms through which a high-ketogenic diet could improve Aβ levels in AD are an 294 alteration in the glycolytic metabolism, a decrease in the generation of ROS, assimilation 295 of ketones as an alternative source of energy in the neurons, and an amelioration in 296 neuronal inflammation 

English is ok, it has only some minor spells

Author Response

> Reviewer2

> 1. Which is the difference between simple summary and abstract?

In accordance with the comment, we modified the simple summary.

> 2. Review does explain many topics but nothing deeply, this must be changed.

Following the advice, we modified the manuscript by adding enhanced information and deeper discussion.

> 3. I think that Diet and AD prevention is a very hot topic, however is explained only shortly in this work.

Following the comment, we added more information on diet and AD prevention (page 8, lines 316-326).

> 4. It will be very interesting to inquiere between other countrys and AD.

It is interesting to inquire between other countries and AD, but we think it is beyond the scope of this paper, in which we focuses on glycolytic metabolism and AD, to discuss the issue. Therefore, we have cited the very recent study on a global epidemiology of dementia including AD (the reference no.1 of the revised manuscript).

> 5. A figure is missing concerning this paragraph: The plausible 293 mechanisms through which a high-ketogenic diet could improve Aβ levels in AD are an 294 alteration in the glycolytic metabolism, a decrease in the generation of ROS, assimilation 295 of ketones as an alternative source of energy in the neurons, and an amelioration in 296 neuronal inflammation 

We added Figure 3 accordingly.

We wish to express our appreciation for the constructive comments, which have helped us significantly improve the manuscript.

Reviewer 3 Report

This present review focus on classic theory of amyloid-β (Aβ) toxicity on Alzheimer Disease and direct to the novel angle of metabolic reprogramming, then leading to the discussion of ketone-mediated nutritional therapeutics, which is fascinating. The review is well written and comprehensive described. However still minor modification can be done to improve.

1. Still can see some typos throughout the texts, for instance, the glycolysis of the simple summary (line4) is mis-spelling. We suggest the authors to check the whole texts again. 2. If any of the figures/diagrams were adapted from previous published version, please provide the resource, for example, the Figure1. 3. Although the main focus of the review lies on metabolic reprogramming, the actual contents seem not that much emphasized on this one, rather the Microbiome-microglia interactions might be heavy part as well. If so, would the authors consider to modify the title in a way that can cover the microbiota part as well, since it will fit the relevance with diet intervention. 4. As for the conclusions and future directions part, would the author consider to make a list of these two parts and similarly, emphasize more on the metabolic reprogramming then diet intervention as potential treatment.

Still can see some typos throughout the texts, for instance, the glycolysis of the simple summary (line4) is mis-spelling. We suggest the authors to check the whole texts again.

Author Response

for Reviewer 3

> This present review focus on classic theory of amyloid-β (Aβ) toxicity on Alzheimer Disease and direct to the novel angle of metabolic reprogramming, then leading to the discussion of ketone-mediated nutritional therapeutics, which is fascinating. The review is well written and comprehensive described. However still minor modification can be done to improve.

Thank you very much for the positive feedback.

> 1. Still can see some typos throughout the texts, for instance, the glycolysis of the simple summary (line4) is mis-spelling. We suggest the authors to check the whole texts again.

We thank the Reviewer3 for carefully reading our manuscript. We have checked the whole text again. We have corrected the misspelling of the word “glycolysis” in Simple Summary. We would be grateful if you could let us know other typos you found.

> 2. If any of the figures/diagrams were adapted from previous published version, please provide the resource, for example, the Figure1.

We confirm that copyright permission is not needed for figures/diagrams of our manuscript. They are completely drawn by the authors without adapting from previously published version.

> 3. Although the main focus of the review lies on metabolic reprogramming, the actual contents seem not that much emphasized on this one, rather the Microbiome-microglia interactions might be heavy part as well. If so, would the authors consider to modify the title in a way that can cover the microbiota part as well, since it will fit the relevance with diet intervention.

We thank the reviewer3 for constructive comments. Accordingly, we modified the title.

> 4. As for the conclusions and future directions part, would the author consider to make a list of these two parts and similarly, emphasize more on the metabolic reprogramming then diet intervention as potential treatment.

According to the advice, we have divided the “Conclusions and future directions” section into two parts; “9. Conclusions” and “10. Future directions” sections. In the re-revised manuscript, we emphasize more on the metabolic reprogramming in the former section, and, then, on the diet intervention as potential treatment in the latter section. We thank the reviewer for the constructive comments, which have helped us significantly improve the manuscript.

Reviewer 4 Report

In this review work, the authors attempted to describe the alterations of aerobic glycolysis in AD.

 The introduction part is too primitive, the main objective and purpose of this review work are missing. It didn't provide any overview of the innovative aspects of the review.

 The description and illustration of metabolic pathways appear to be a replica of existing information. The article seems like reading a normal textbook overloaded with basic information.

The overall scientific debate, interpretation of existing reports, and description of available new literature are too weak.

 Many statements in the manuscript are repeated which dilutes the readability and interest.

 The supportive reference appears to be missing for many statements.

 The article focuses on many irrelevant topics and failed to establish a new

 concept or hypothesis. 

It is fine 

Author Response

for Reviewer 4

> In this review work, the authors attempted to describe the alterations of aerobic glycolysis in AD.

 > The introduction part is too primitive, the main objective and purpose of this review work are missing. It didn't provide any overview of the innovative aspects of the review.

The main objective and purpose of this review work is to describe the emerging features of Alzheimer’s disease from the aspect of alternation in the central metabolic pathways of carbohydrate metabolism in the human brain. This has been written in the introduction section (lines 52-53). The innovative aspect of this review is focusing on the metabolic alternation, i.e., metabolic reprogramming toward aerobic glycolysis, involved in the brain amyloid pathology. This has been written in the introduction section (lines 49-51). We added the statement, “possible dietary intervention for AD including ketone-mediated nutritional therapeutics is also discussed,” at the end of the introduction section (lines 53-54).

Round 2

Reviewer 1 Report

All comments were responded by the authors. 

Author Response

Thank you so much for the kind response.

Reviewer 2 Report

With the new changes now is correct, it is ready to publish.

Author Response

Thank you so much for the good response.